# A Sub-Seasonal Crop Information Identification Framework for Crop Rotation Mapping in Smallholder Farming Areas with Time Series Sentinel-2 Imagery

**Huaqiao Xing [1], Bingyao Chen [1,*] and Miao Lu [2]**

[1]    School of Surveying and Geo-Informatics, Shandong Jianzhu University, Jinan 250101, China
[2]    Key Laboratory of Agricultural Remote Sensing, Ministry of Agriculture and Rural Affairs/Institute of
       Agricultural Resources and Regional Planning, Chinese Academy of Agricultural Sciences,
       Beijing 100081, China
*      Correspondence: 2020165108@stu.sdjzu.edu.cn

**Abstract:** Accurate crop rotation information is essential for understanding food supply, cropland management, and resource allocation, especially in the context of China's basic situation of "small farmers in a big country". However, crop rotation mapping for smallholder agriculture systems remains challenging due to the diversity of crop types, complex cropping practices, and fragmented cropland. This research established a sub-seasonal crop information identification framework for crop rotation mapping based on time series Sentinel-2 imagery. The framework designed separate identification models based on the different growth seasons of crops to reduce interclass similarity caused by the same crops in a certain growing season. Features were selected separately according to crops characteristics, and finally explored rotations between them to generate the crop rotation map. This framework was evaluated in the study area of Shandong Province, China, a mix of single-cropping and double-cropping smallholder area. The accuracy assessment showed that the two crop maps achieved an overall accuracy of 0.93 and 0.85 with a Kappa coefficient of 0.86 and 0.80, respectively. The results showed that crop rotation practice mainly occurred in the plains of Shandong, and the predominant crop rotation pattern was wheat and maize. In addition, Land Surface Water Index (LSWI), Soil-Adjusted Vegetation Index (SAVI), Green Chlorophyll Vegetation Index (GCVI), red-edge, and other spectral bands during the peak growing season enabled better performance in crop mapping. This research demonstrated the capability of the framework to identify crop rotation patterns and the potential of the multi-temporal Sentinel-2 for crop rotation mapping under smallholder agriculture system.

**Keywords:** crop rotation mapping; a sub-seasonal framework; smallholder agriculture; feature selection; time series; Sentinel-2; Google Earth Engine



## 1. Introduction

"Small farmers in a big country" is an accurate portrayal of China's agricultural situation. According to data from the National Bureau of Statistics' Third National Agricultural Census [1], smallholder farmers operate 70% of the total cropland area. Crop mapping in smallholder agriculture is essential for understanding food supply, cropland management, and resource assignment [2,3].

The availability of Sentinel-2 data has provided better support for crop mapping, as its unique three red-edge bands are particularly beneficial for crop mapping [3]. Sentinel-2 imagery is now being used successfully for crop mapping at various scales in different regions, such as the county scale of the USA and Nigeria [4,5], the provincial scale of China [6–8], the state scale of Brazil [9], as well as national scales [10], etc. Despite these great advances on crop mapping, they are mostly concentrated on the identification of crop types in single-cropping areas, with less research on crop rotations in double-cropping

areas. Crop rotation means that different crops are grown in seasonal sequence on the same plot, resulting in a double-cropping or multi-cropping system [11]. Existing studies on crop rotation have either examined the cropping patterns of a particular crop (e.g., rice) [12], mapped specific rotation types within a year [13,14], or have focused on the variation in crop types grown between adjacent years [15]. However, studies of crop rotation in smallholder areas with complex cropping systems (mix of single-cropping and double-cropping) have not been fully investigated.

Different from most other land cover types (i.e., forest, grass, and wetland), crops undergo a full growth cycle from sowing to harvest in a relatively short period of time [15,16]. Each crop exhibits unique characteristics at different stages of growth, known as phenological information, which is the key basis for crop rotation mapping. However, crop phenology also greatly varies spatially, as it depends on many factors, e.g., cropping practices, topography, soil quality, and climatic conditions [15]. These factors can lead to high intraclass variability in the spectro-temporal signal and increase with the complexity of cropping patterns [7]. Therefore, crop rotation mapping based on remote sensing remains challenging due to the complex cropping practices, the spatial and temporal variability of crop types, and the fragmented smallholdings [17]. Traditional methods of extracting crop rotation information tend to define the crop rotation type directly and map each rotation type as a class [14,18,19]. These approaches not only require extensive auxiliary data and expert knowledge to determine specific rotation types, but also require the analysis of images throughout the crop year. The problem of interclass similarity between different crop rotation types when there is one season of the same crop in double-cropping pattern may result in classification errors.

To address the above challenge, this study proposed a sub-seasonal crop information identification framework for crop rotation mapping within a year. The framework designed separate identification models based on the different growing seasons of crops and selected features separately according to the crop characteristics to ensure the accuracy of crop mapping. The crop rotation map was then mapped based on spatial and attribute analysis of the crop maps. This framework can minimize the uncertainties caused by direct mapping of crop rotation types. This study explored the possibility of the framework using Shandong Province as the study area. Intercropping and crop rotation are common in the study area, and both single-cropping and double-cropping crops are grown [20]. The main crops in the study area were firstly classified into summer harvest and autumn harvest crops according to their growing season. Then, different identification algorithms were designed based on the characteristics of crops to generate two separate crop type maps. The final crop rotation map for Shandong was generated by combining the crop information from the two crop type maps.

## 2. Materials and Methods

### 2.1. Study Area

Shandong Province is located on the eastern coast of China and downstream of the Yellow River (between 34°22′–38°15′N and 114°19′–122°43′E) (Figure 1). The terrain of Shandong is predominantly plain or mountainous hilly areas with complex topography [21]. The western and northern parts belong to the North China Plain, the mountainous and hilly areas are located in the south-central part of the country, while the eastern part is a peninsular region bordering the sea [22,23]. Shandong is situated in the warm temperate zone, with a suitable climate, abundant sunshine, and concentrated precipitation [24]. According to the Shandong Statistical Yearbook 2021 (Natural Resources 2020), cropland accounts for about 48% of the province's total land area [25]. The vast area of cropland makes it an important grain production base in China [20]. Based on the climatic and topographical conditions, the cropping system in Shandong can be categorized into two types: single-cropping crops and double-cropping crops. Moreover, according to the Shandong Statistical Yearbook and field surveys, there are two types of crops in Shandong: summer harvest (mainly wheat) and autumn harvest (maize, rice, peanuts, cotton etc.).

Wheat and maize are the primary grain crops, accounting for 71.7% of the province's total sown area of farm crops, and they are usually grown in crop rotation. In addition, peanuts and cotton are the main cash crops in Shandong. Referring to the categorization of crops in the National Statistical Yearbook, this research focused on the summer harvest crop of wheat and the autumn harvest crops of maize, rice, peanuts, and cotton [25].

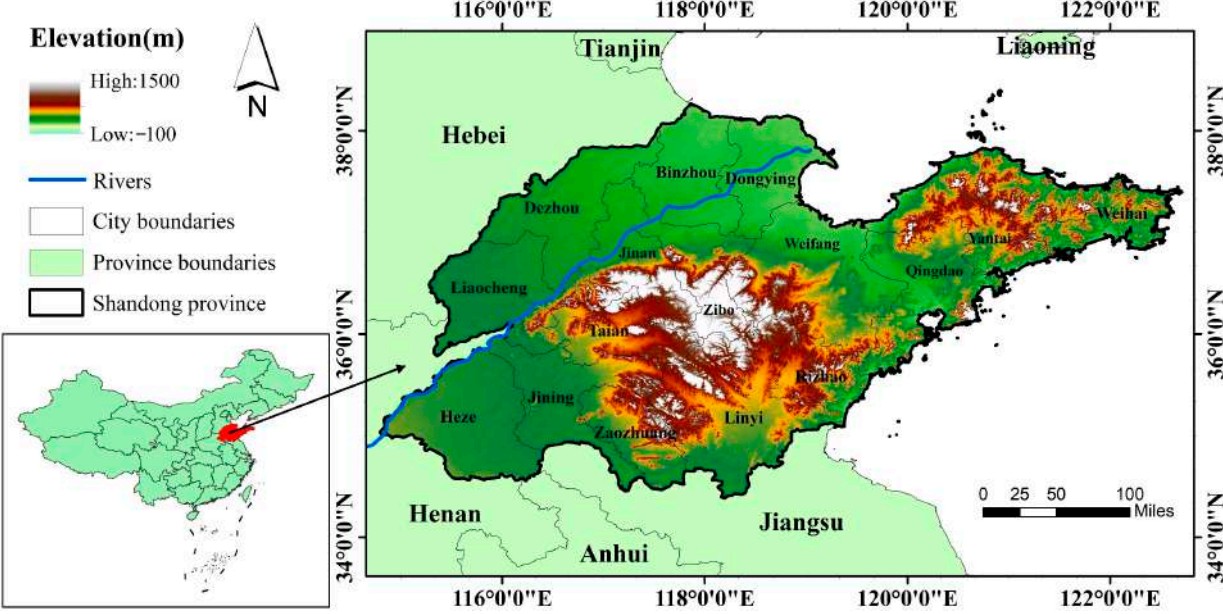

**Figure 1.** The geographical location and topography of Shandong Province.

*2.2. Datasets and Pre-Processing*

2.2.1. Satellite Data and Pre-Processing

Sentinel-2 is an earth observation mission under the Copernicus program of the European Space Agency and is a constellation of two identical satellites, Sentinel-2A/B [26]. It has been successfully used in many crop identification studies because of its high spatial and temporal resolution and, in particular, the special contribution of the red-edge bands to crop mapping [3,10].

A crop year was defined by the growth cycle of all crops in a year and usually ranges from the planting date of the first crop to the harvest date of the last crop [27]. For example, the crop year 2020 in Shandong is from October 2019 to October 2020. A total of 2687 scenes of Sentinel-2 Level-2A images with less than 50% cloud cover in the 2020 crop year were accessed from the Google Earth Engine (GEE) catalogue "COPERNICUS/S2_SR". It was also de-clouded using the QA60 band, which is dedicated to providing information on cloud status, to exclude clouds and other bad-quality observations. The rest of the pixels were reserved as high-quality observations [28], so that each pixel at each location has a number of high-quality observations. The spatial distribution of the high-quality observation number per individual pixel is shown in the Figure 2, with over 90% of individual pixels having more than 40 high-quality observations. On this basis, monthly image collection (13 in total) was composited to characterize crops phenology, and gaps were filled in with images from the months of the adjacent years [29]. Ten bands of Sentinel-2 images were selected in the classification process, these are B2, B3, B4, B5, B6, B7, B8, B8A, B11, and B12. The six bands with a spatial resolution of 20 m were resampled to 10 m by the nearest neighbor resampling method to maintain the same spatial resolution.

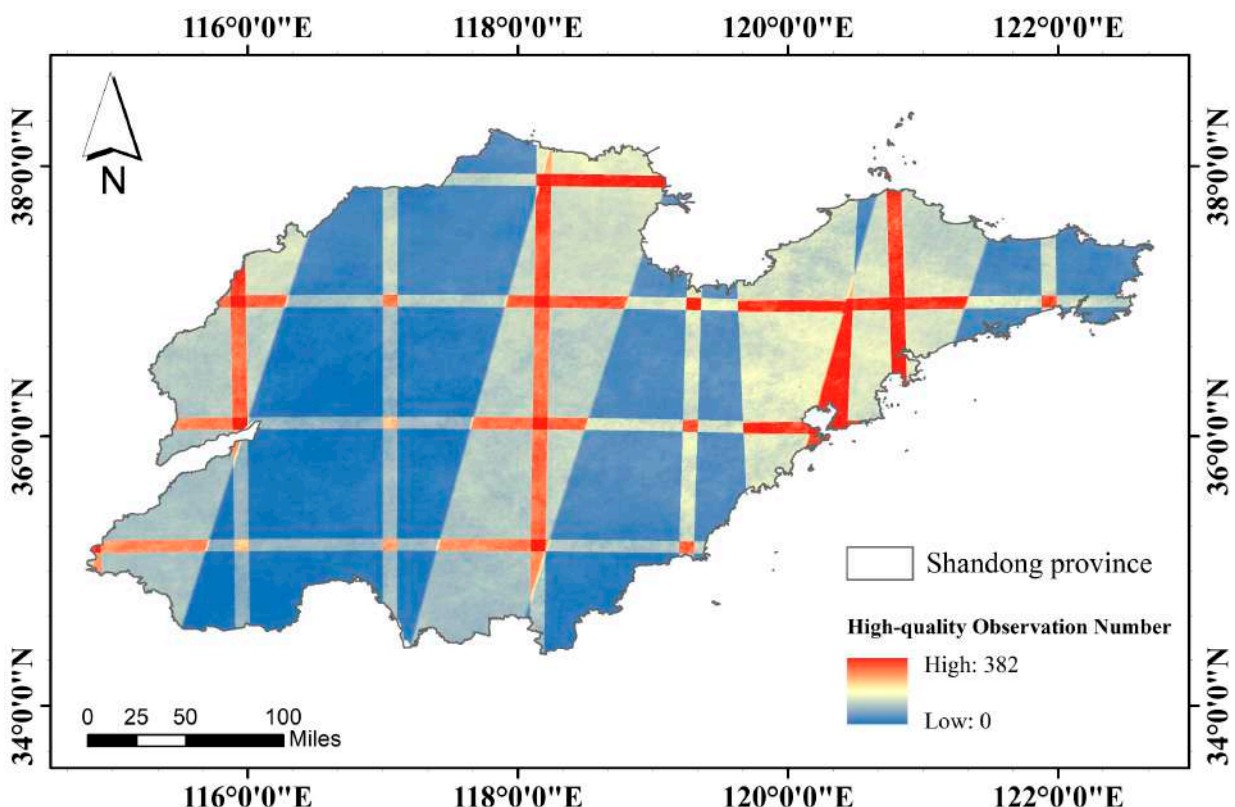

**Figure 2.** The spatial distribution of the high-quality observation number in Shandong during the 2020 crop year.

### 2.2.2. Ground Reference Data

The reference dataset was constructed to train and validate the algorithms proposed in this research. It was generated through ground surveys and the interpretation of Google Earth images. Sample points of homogeneous plots with similar color and texture to the ground survey points during the same period were added to the reference data collection (Figure 3). The reference dataset contains a total of 1261 crop sample points (Table 1), of which 70% were used for model training and the other 30% for validation. As shown in Table 1, the distribution of samples across crop types is imbalanced, which is caused by the uneven distribution of land cover. This research was fully informed by the information provided by the China Agricultural Information Network [30] on the sowing/harvest time, growth status, and the phenological periods of different crops, and the crops calendar obtained is shown in Figure 4.

### 2.2.3. Ancillary Data

Other ancillary data used in this study include the Shuttle Radar Topography Mission (SRTM), which is an international research effort that has resulted in a near-global digital elevation model with a resolution of 30 m [31]. It was used to calculate elevations and slopes to explore the effects of topography on crops. The SRTM data are available in GEE from the data catalogue "USGS/SRTMGL1_003".

### 2.3. Methodology

Based on time-series Sentinel-2 imagery, this research proposed a sub-seasonal crop information identification framework for crop rotation mapping under complex cropping patterns (Figure 5). The workflow of the study includes the following steps: (a) construction of a high-quality monthly image collection; (b) feature selection; (c) mapping of crop types and crop rotation; and (d) accuracy assessment. The detailed steps are described in the following sections.

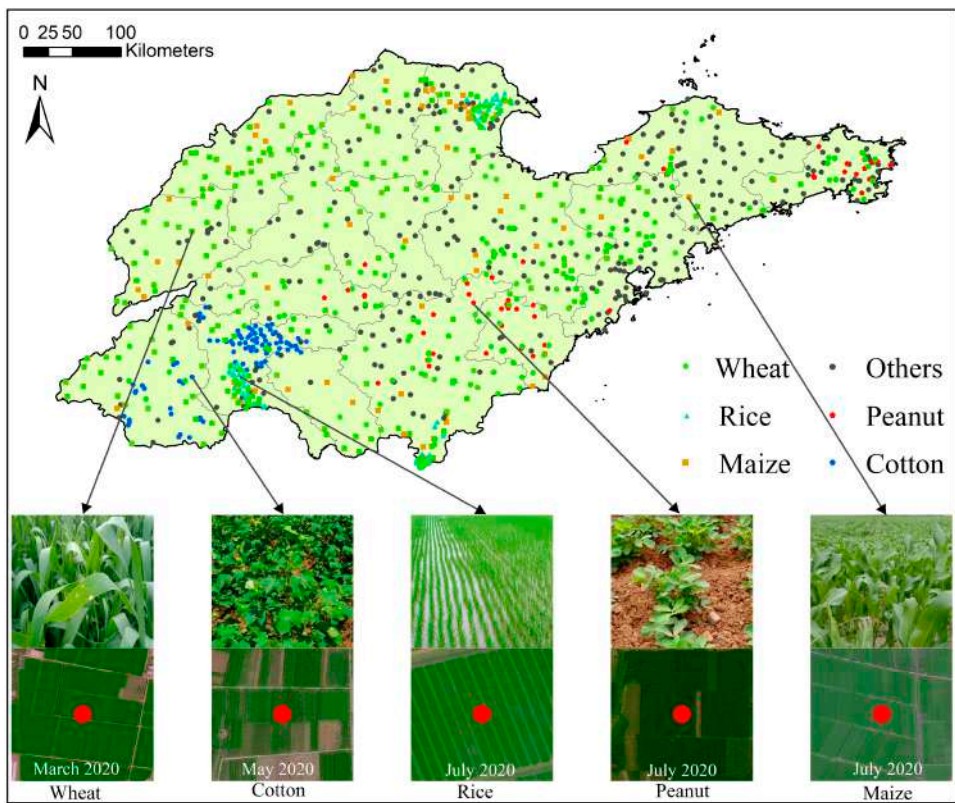

**Figure 3.** The spatial distribution of samples and the real growth status of each crop at two growing seasons (May or July 2020) as well as the corresponding Google Earth images.

**Table 1.** Properties and specific quantities of the reference sample dataset.

| Category | Wheat | Maize | Rice | Cotton | Peanut | Others | Total |
|---|---|---|---|---|---|---|---|
| Training | 267 | 193 | 65 | 61 | 74 | 223 | 883 |
| Validation | 114 | 83 | 28 | 26 | 32 | 96 | 378 |

**Figure 4.** Agricultural calendar of main crops in Shandong.

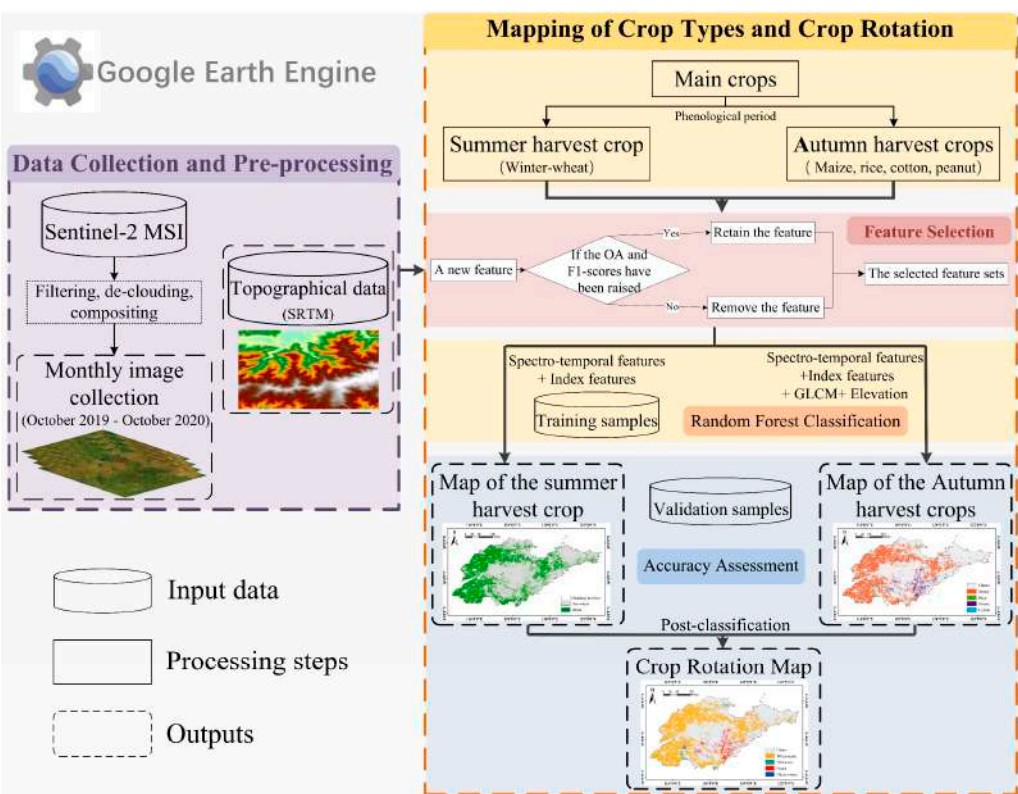

**Figure 5.** The workflow for crop rotation mapping in this study.

### 2.3.1. Feature Selection

Crop rotation is a common practice in the study area [32]. The summer harvest crop is mainly wheat, while the autumn harvest crops are more complex, with maize, rice, cotton, and peanuts. More importantly, there are significant differences in the seeding and harvesting date of the above crop types. To address the complex crop rotation pattern, two separate crop identification systems were developed for extracting the summer harvest crop and the autumn harvest crops, respectively.

Both identification systems are based on spectro-temporal features, that is, reflectance in 10 bands (B2, B3, B4, B5, B6, B7, B8, B8A, B11, B12) of the monthly images. On this basis, different features were added to better express the variability between different crops (i.e., wheat and others). Other features were selected by firstly considering the vegetation indices commonly used in previous studies and secondly by assessing the Overall Accuracy (OA) and the F1-scores of the main crop types throughout the growing season. If the added feature index improved the OA or made a unique contribution to the identification of a crop type, it was retained in the system, otherwise it was removed. As shown in Table 2, the following indices were added to the wheat identification system: Normalized Difference Water Index (NDWI), Green Chlorophyll Vegetation Index (GCVI), Land Surface Water Index (LSWI), Soil-Adjusted Vegetation Index (SAVI), and Normalized Differential Phenology Index (NDPI). The Normalized Difference Vegetation Index (NDVI), normalized difference residue index (NDRI), NDWI, GCVI, LSWI, eight texture features (mean, entropy, contrast, correlation, variance, dissimilarity, asm, homogeneity) under the grey-level co-occurrence matrix (GLCM) and elevation obtained from SRTM were added to the autumn harvest crop identification system.

**Table 2.** The spectral indices selected in this research and their expressions.

| Indicators | Expressions | References |
|:---:|:---:|:---:|
| NDVI | $NDVI = \frac{\rho_{NIR}-\rho_R}{\rho_{NIR}+\rho_R}$ | [33] |
| NDWI | $NDWI = \frac{\rho_G-\rho_{NIR}}{\rho_G+\rho_{NIR}}$ | [34] |
| NDRI | $NDRI = \frac{\rho_{SWIR1}-\rho_R}{\rho_{SWIR1}+\rho_R}$ | [35] |
| GCVI | $GCVI = \frac{\rho_{NIR}}{\rho_G} - 1$ | [36] |
| LSWI | $LSWI = \frac{\rho_{NIR}-\rho_{SWIR2}}{\rho_{NIR}+\rho_{SWIR2}}$ | [37] |
| SAVI | $SAVI = \frac{(\rho_{NIR}-\rho_R)}{(\rho_{NIR}+\rho_R+0.5)} \times (1+0.5)$ | [38] |
| NDPI | $NDPI = \frac{\rho_{NIR}-(0.74\times\rho_R+0.26\times\rho_{SWIR1})}{\rho_{NIR}+(0.74\times\rho_R+0.26\times\rho_{SWIR1})}$ | [39] |

Note: $\rho_G$, $\rho_R$, $\rho_{NIR}$, $\rho_{SWIR1}$, and $\rho_{SWIR2}$ are the reflectance values of Sentinel-2 bands 3, 4, 8, 11,12.

### 2.3.2. Crop Type Identification and Rotation Mapping

This research used the Random Forest (RF) algorithm to identify different types of crops at first. RF is a robust voting decision algorithm by integrating multiple independent decision trees, and it is widely used for land cover classification in previous studies [14,40,41]. This algorithm is now well-integrated into the GEE platform with the classification library of "ee.Classifier.smileRandomForest()". The number of decision trees was determined by constantly testing to achieve higher accuracy and classification efficiency, with the other parameters were set to default values [12].

The vast majority of cropland in Shandong is planted with wheat in the first growing season, so only wheat was extracted for the summer harvest crop. In the study area, wheat is generally sown in October of the previous year and harvested in June of the current year. Thus, monthly images covering the entire growth period of wheat (October 2019–June 2020) were selected for wheat mapping. The 10 spectral bands with 5 indices (NDWI, GI, SAVI, LSWI, NDPI) formed a spectro-temporal feature set with a total of 135 features. Those features were input into the RF classifier and the number of decision trees was set to 200 after repeated experiments, resulting in an initial summer harvest crop map.

Monthly images from April to October 2020 were used to map the four crop types in the second growing season. In addition to the temporal feature set of 10 spectral and 5 effective index features (NDVI, NDRI, NDWI, GI, LSWI) from monthly image collection, as well as texture features and elevation were added for the four crop types mapping, resulting in a feature set with a total of 114 variables. The features set was input into the classification system for the autumn harvest crops and the number of decision trees was set to 460 after repeated experiments, resulting in an initial autumn harvest crops map.

Spatial overlay analysis of the two crop maps showed that over 96% of maize and 91% of rice coincided with wheat spatially; however, less than 1% of cotton plots matched with wheat plots. This is sufficient to indicate that maize and rice were sown after wheat, while cotton was not. This, combined with the analysis of NDVI time series (Figure 6), the two crop maps were integrated to five classes: wheat–maize, wheat–rice, peanut, other crops–cotton and others to generate the crop rotation map.

It is a remarkable fact that the results of pixel-based RF classifiers will inevitably produce the "salt and pepper", which means that a large homogeneous area of one class may contains a single or small number of misclassified pixels of another class [29]. This is particularly problematic for the smaller plots of cropland under smallholder agriculture. Therefore, the crop rotation map needs to be post-processed to remove classification noise and enhance the reliability. A post classification smoother is available in GEE, but it has the disadvantage that fine features such as rivers and roads are obscured in the post-processed maps [29]. This research proposed an object-based approach to post-processing pixel-based mapping results. Firstly, this study obtained image objects through the image segmentation algorithm provided by GEE. The pixel-based crop rotation results were then optimized based on the object layers using the majority voting scheme, which has been used in previous studies [42,43]. Then, the final crop rotation map for Shandong for 2020 was generated.

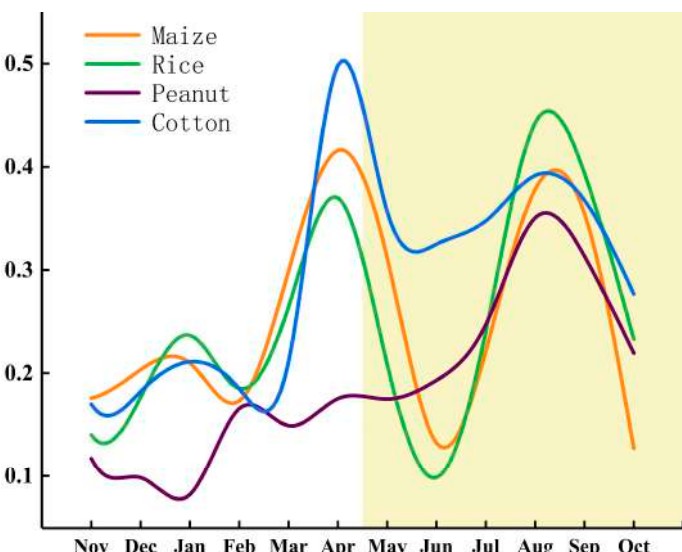

**Figure 6.** Averaged NDVI time profiles of different crops throughout the crop year (the shaded area is the autumn harvest crops growth period).

2.3.3. Accuracy Assessment

This research evaluated the accuracy of the mapping results in two ways. Firstly, the generated spatial distribution maps of wheat and autumn harvest crops were evaluated using the independent validation samples. Four metrics derived from the confusion matrix, OA, User's Accuracy (UA), Producer's Accuracy (PA), Kappa coefficient, and F1-score (Equation (1)), were used to comprehensively describe the classification accuracy of each crop type [26]. In addition, the accuracy of the crop rotation map was evaluated in some test areas using ultra-high resolution Google historical imagery, as no data on the spatial distribution of crop rotation information were available.

$$F1 - score\ = 2 \times \frac{PA \times UA}{PA + UA} \tag{1}$$

**3. Results**

*3.1. Spectro-Temporal Features for Crop Rotation Mapping*

The cropping intensity can often be determined by the number of peaks in the temporal NDVI profile [12,44]. In general, there is one peak for the single-cropping system and two peaks for the double-cropping system. Figure 6 shows mean temporal NDVI profiles throughout the crop year based on the autumn harvest crops samples to analyze the crop rotation. It can be seen from Figure 6 that all crops except peanut have two distinct NDVI peaks. It is reasonable to assume that maize, rice, and cotton belong to the double-cropping system, while peanut is a single-cropping pattern. The NDVI curves for maize and rice are very similar, with a total of three peaks around December, April, and August. The last two are distinct peaks with higher NDVI values, representing the peak growing seasons of the first and second season crops, respectively, while the first peak is closely related to the first crop (wheat). This is because the NDVI curve of wheat generally shows a distinctive pattern with two peaks at tillering and heading stages [7]. In addition, the first NDVI peak in April and the NDVI values in June and July of cotton is much higher than that of other crops in the same period. After investigation, there are two types of double-cropping cotton fields in Shandong: seedling transplants and direct sowing fields. The seedling transplanting cotton fields is transplanted after seedlings in greenhouses intercropping with garlic, onions, or other small crops etc. The direct sowing cotton fields is to sow short-season cotton after the harvest of garlic, potatoes, onions, or other crops, and must ensure that all seedlings by June 10. Unlike other crop rotations, cotton is already growing in June, while wheat is

maturing for harvest and the other crops in the rotation with wheat have not been sown, so the NDVI for cotton will be much higher than the NDVI for other crops during this period.

Figure 7 shows the averaged spectral reflectance of the different crops by month. In general, the differences in reflectance of autumn harvest crops are most significant in May, June, and July, especially in the red-edge and near infrared (NIR) bands (B6, B7, B8, B8A). As shown, the reflectance of peanut in April is significantly different from that of the other crops. This is because peanut is in a newly sown or unseeded state in April. Additionally, peanut is a single-cropping crop, which is clearly different from other rotational crops that are in the first peak of the growing season, so it is more easily distinguished from other crop types. In contrast, cotton has significantly higher reflectance in the red-edge and NIR bands than other crops in May, June, and July. Both maize and peanut are significantly different from the other crops in the red-edge and NIR bands in October. Additionally, because peanut has distinctive features that are clearly different from other crops, maize can be identified using the red-edge and NIR bands in October. Additionally, the red-edge reflectance in May, June, July, and September are important features to distinguish rice from other crops.

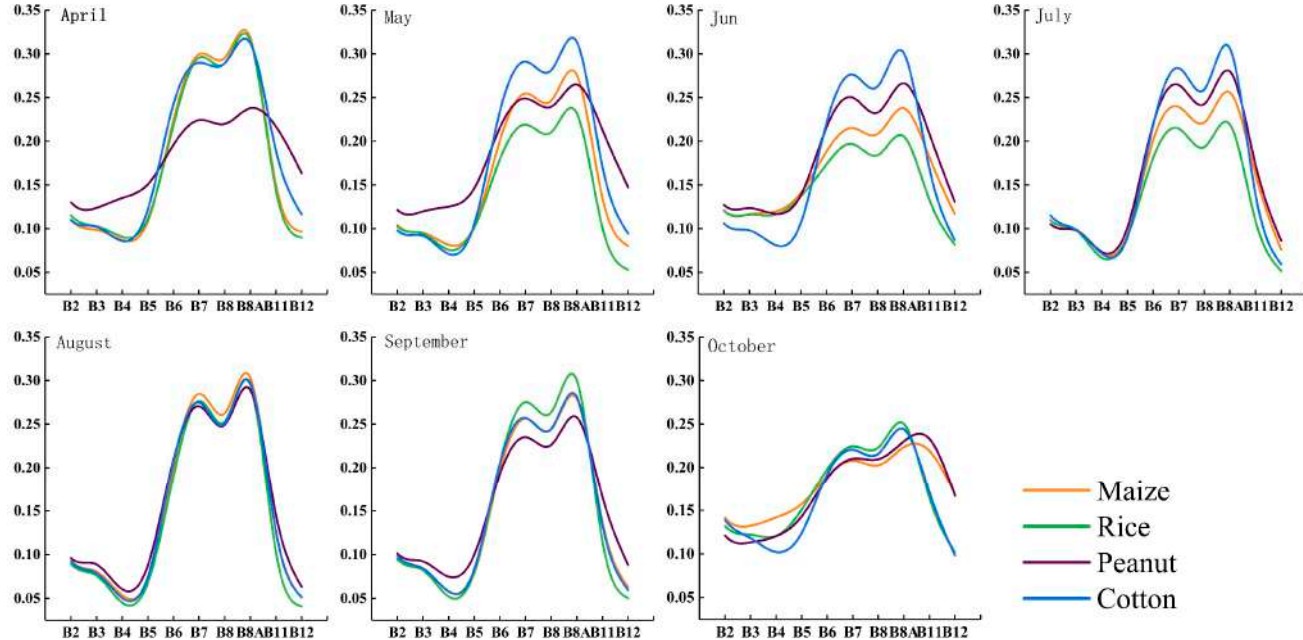

**Figure 7.** Differences between the averaged spectral reflectance of Sentinel-2 for autumn harvest crops in different months.

Figure 8 shows the temporal variation in the mean values of the different vegetation indices of crops. As shown in this figure, the mean GCVI, NDRI, and NDVI values of cotton in April are much higher than those of the other three crops, and the NDVI values in May, June, and July as well as the NDWI and LSWI values in June are significantly different from those of the other crops. The mean GCVI and NDVI values of maize in October are slightly lower than those of the other crops, especially the LSWI values of maize and peanut in June, July, and October are significantly different from the other crops, which are very important for the identification of maize. In contrast, the LSWI values for peanut are significantly lower than those of others for almost the entire growing season, which shows the importance of LSWI for peanut identification. The NDRI values of rice are lower than those of other crops throughout the growing season, especially in May, June, and July. Additionally, the LSWI values in May, August, and September are also effective in distinguishing rice from others. In general, LSWI and NDRI in May, June, July, and August are key features to distinguish between the various crops.

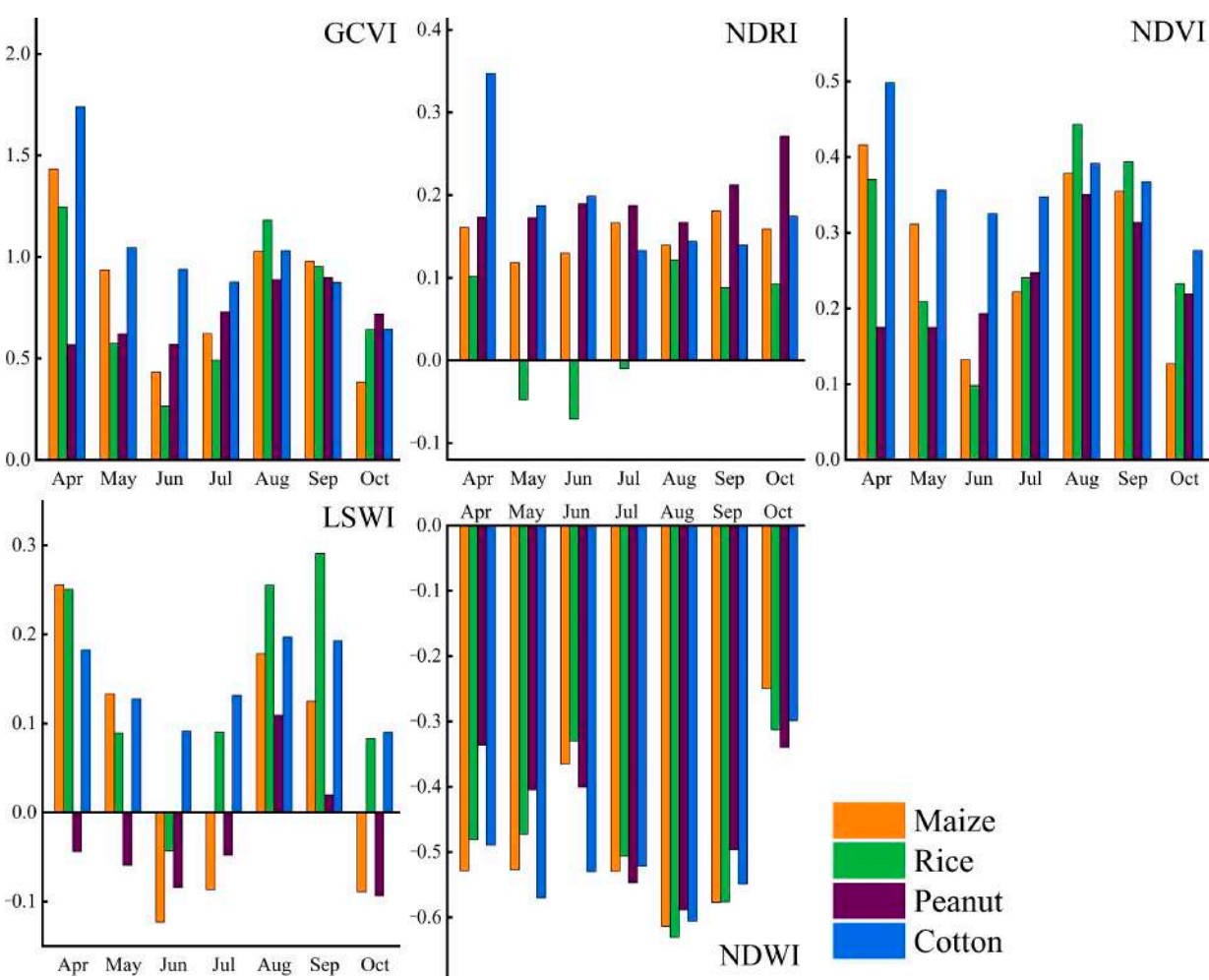

**Figure 8.** Temporal differences in the mean values of the vegetation indices for autumn harvest crops.

### 3.2. The Spatial Patterns of Crop Types and Rotation

This research extracted the spatial distribution of summer harvest and autumn harvest crops in Shandong based on Sentinel-2 time series data, and then generated a 10-m resolution crop rotation map. As shown in Figure 9, the summer harvest crop of wheat is widely distributed in the western, northern, and eastern plains of Shandong. The autumn harvest crop (Figure 10), on the other hand, is dominated by maize, whose spatial distribution is roughly the same as that of wheat. Rice has three main planting areas, south-west of Jining, south of Linyi, and north of Dongying, and all of them have good water conditions. Peanut is most scattered and is mainly distributed in the hilly areas of south-central Shandong. In addition, cotton is grown on a relatively small area, mainly in the northern part of Jining. The crop rotation map (Figure 11) shows that crop rotation is mainly found in the plains, with wheat in the first growing season and maize in the second being the main crop rotation pattern. Large areas of wheat–maize are concentrated in the northwestern plain, the southwestern plain, and the Jiaolai plain of Shandong, especially in Heze, Jining, Liaocheng, and Dezhou. Rice is mostly planted after the wheat harvest, forming a wheat–rice crop rotation. As previously analyzed, peanut is a single-cropping pattern excluding rotation information. The analysis of the time-series NDVI shows that cotton is the second-season crop in Shandong, and the first-season crop is not wheat but other small crops which are not included in this study, so it is mapped separately as "Others–cotton" in the crop rotation map.

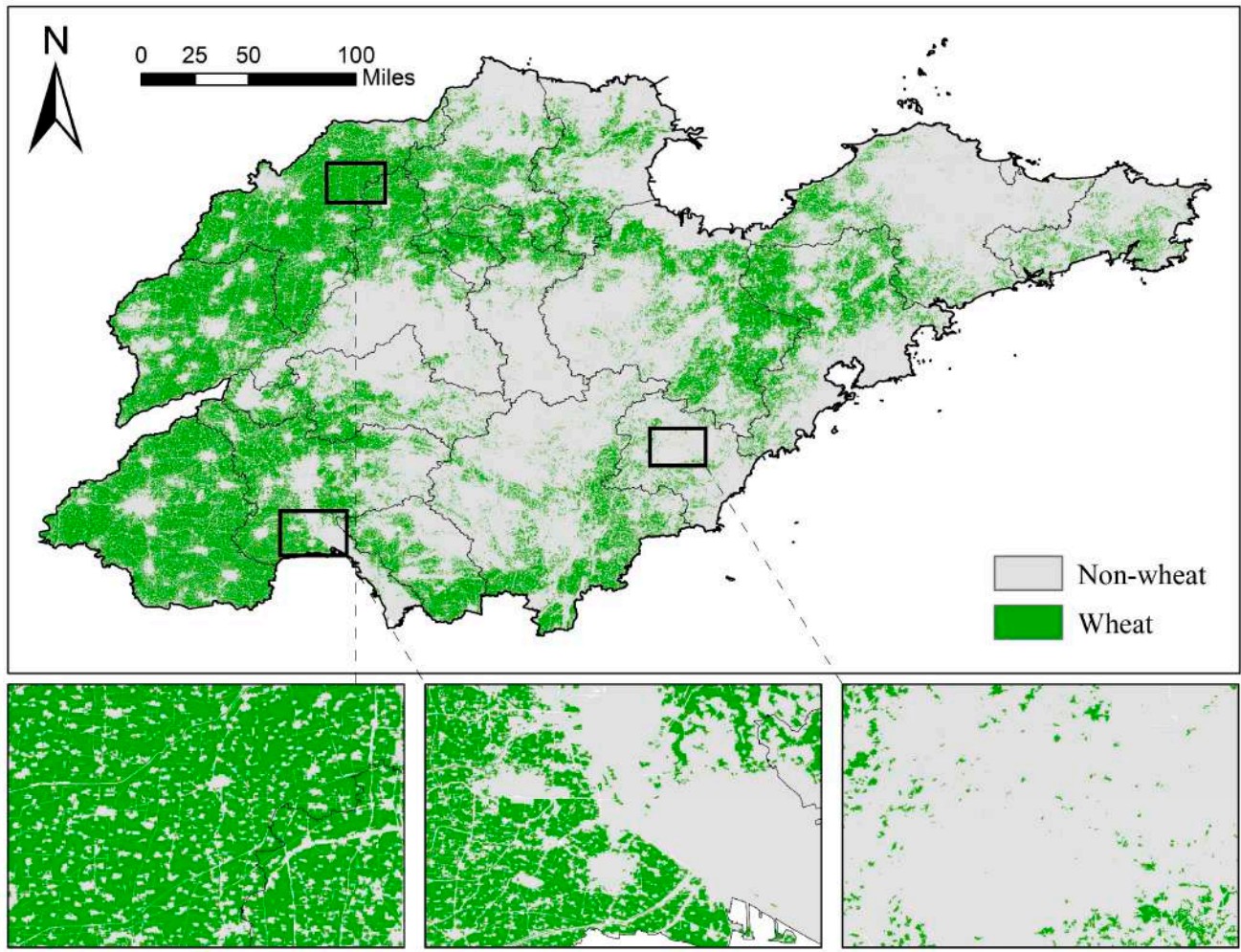

**Figure 9.** Spatial distribution and the close-up views of the summer harvest crop (winter-wheat) in Shandong Province.

### 3.3. Accuracy Assessment of Crop Maps

The accuracy of the two crop maps was assessed using validation samples. The results showed that the OA of the summer harvest crop map was 0.93 with a Kappa coefficient of 0.86, and the OA of the autumn harvest crop map was 0.85 with a Kappa of 0.80 (Table 3). In the autumn harvest crop map, maize has the highest recognition accuracy, with an F1 score of 0.86, while peanut has the lowest accuracy, with an F1-score of 0.73. Peanut is mostly grown in hilly areas, where the highly fragmented cropland produces more mixed pixels, resulting in lower accuracy of peanut mapping [27].

**Table 3.** Confusion matrix of autumn harvest crops identification.

| Classification Map | Reference Samples | | | | | | | |
|---|---|---|---|---|---|---|---|---|
| | **Others** | **Maize** | **Rice** | **Peanut** | **Cotton** | **Total** | **UA (%)** | **F1-Score** |
| Others | 94 | 0 | 1 | 4 | 1 | 100 | 0.94 | 0.88 |
| Maize | 8 | 75 | 2 | 2 | 0 | 87 | 0.86 | 0.86 |
| Rice | 2 | 5 | 25 | 0 | 0 | 32 | 0.78 | 0.83 |
| Peanut | 6 | 3 | 0 | 23 | 1 | 33 | 0.70 | 0.73 |
| Cotton | 3 | 4 | 0 | 1 | 29 | 37 | 0.78 | 0.85 |
| Total | 113 | 87 | 28 | 30 | 31 | 289 | OA = 0.85 | |
| PA (%) | 0.83 | 0.86 | 0.89 | 0.77 | 0.94 | | Kappa = 0.80 | |

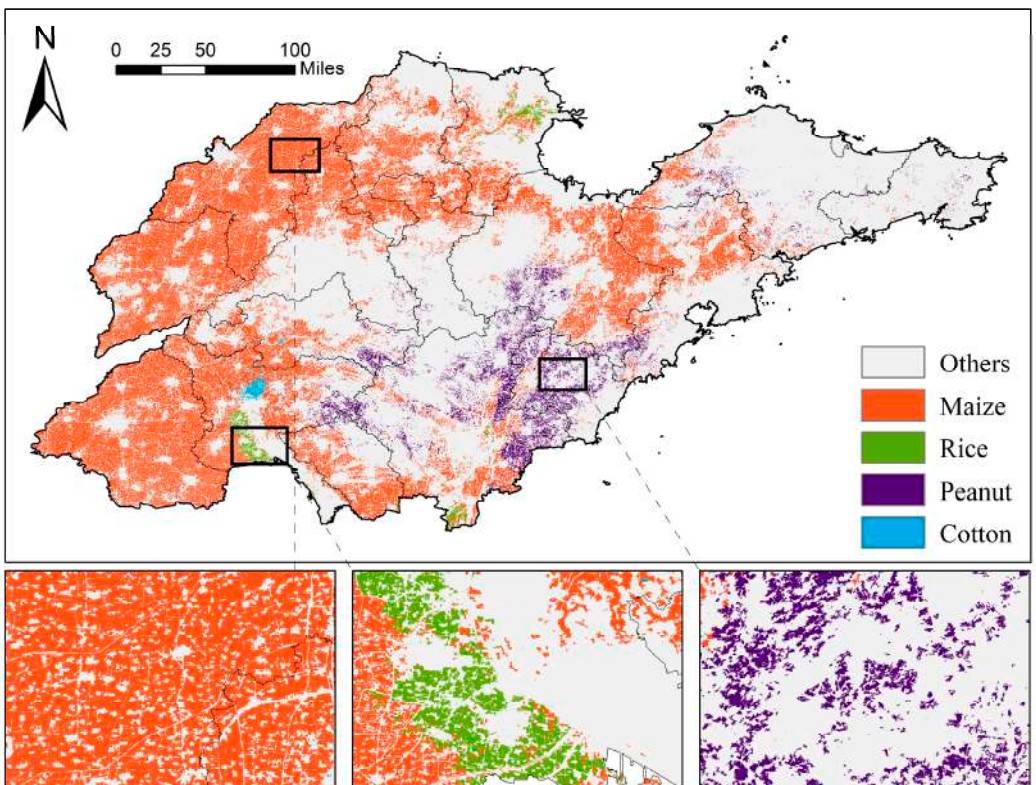

**Figure 10.** Spatial distribution and the close-up views of the autumn harvest crops (maize, rice, peanut, and cotton) in Shandong Province.

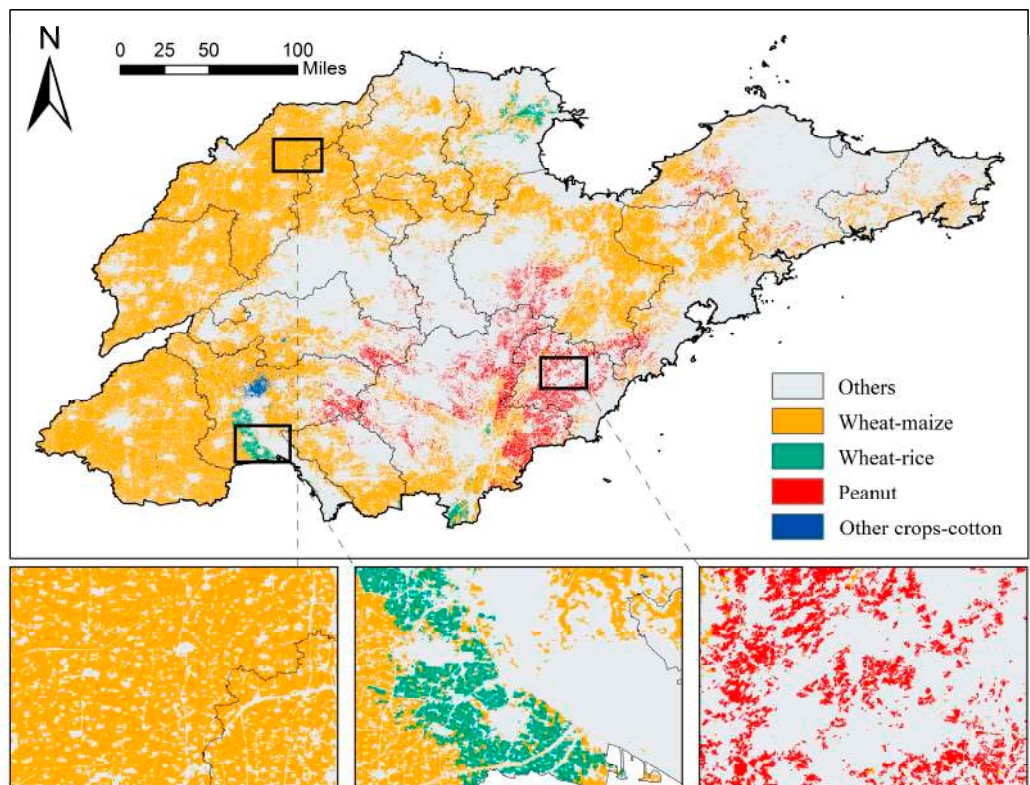

**Figure 11.** Crop rotation map with 10 m resolution and close-up views of Shandong Province in 2020.

The visual evaluation showed that the crop rotation map obtained in this study achieved a high level of agreement with the interpretation of Google historical imagery [45].

## 4. Discussion

### 4.1. Potential of Time Series Images for Crop Rotation Mapping

This research explored the role of Sentinel-2 images in early crop identification by performing an iterative RF classification by adding images month by month. First, the earliest monthly images within the crops growing season were used separately for classification, and the F1-score was used to assess the classification accuracy of each crop. Secondly, the monthly images closest in time to the image undergoing classification were added to the classifier one by one to perform classification and accuracy evaluation based on the same training and testing samples, respectively.

As shown in Figure 12a, the OA of 81.3% and F1-score of 0.84 were achieved with only one image of October during the wheat mapping process. After adding images of November and December one by one, the OA reached 87.3% and the F1-score was 0.85, and then it remained relatively stable until March of the following year. With the addition of the April image of the current year, the OA exceeded 90% for the first time and the F1-score reached 0.90. This is because the previous year, from late October to December, is the tillering period for wheat. Unlike other vegetation covers, wheat has a unique growth period during this time [46], which is an important reason for the high accuracy of wheat identification. The peak growing season (jointing and heading dates) starts in March and April of the following year [7]. With the addition of the April image, the identification accuracy of wheat was further improved, ending up with an OA of 94.6% with an F1-score of 0.92.

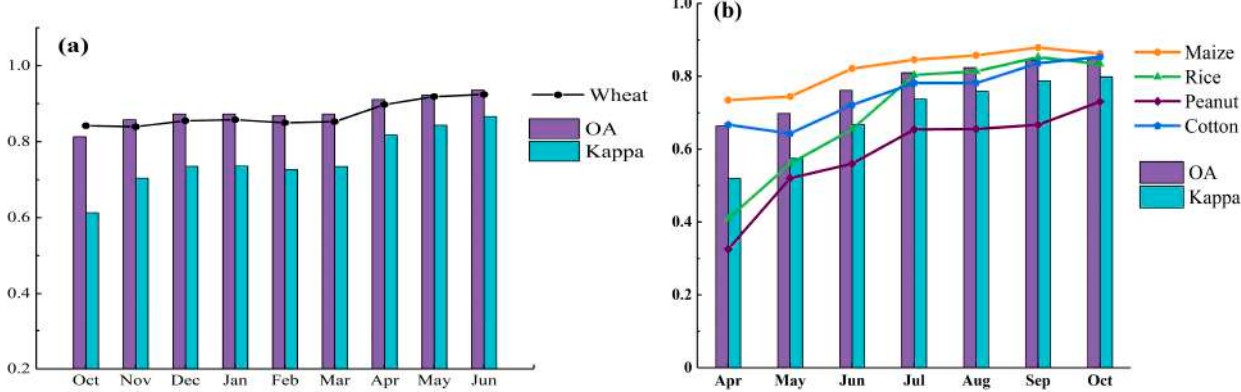

**Figure 12.** Overall accuracy, Kappa coefficient, and F1-score (polyline) for wheat mapping (**a**) and autumn harvest crops mapping (**b**) at monthly time step.

Figure 12b shows the change in F1-scores for the four types of autumn harvest crops as well as the OA and Kappa of mapping with the increasing number of multi-temporal images. It can be seen that the OA increased with the increasing number of temporal images until August, and then the growth remains small and steady. After the addition of the July image, the OA exceeded 80% with a Kappa coefficient of 0.74. Correspondingly, the F1-score for rice stabilized after this time point when it exceeded 0.8. This indicates that the spatial distribution of rice can be identified using the information of all time-series images before July. For maize, the F1-score peaked after the addition of the September image (F1 = 0.88), while it decreased by 0.02 after the addition of the October image, as was the case for rice. This means that for both maize and rice, the additional October image may have introduced redundant information, which negatively affected the classification system [47]. The F1-scores for peanut and cotton reached their maximum after the addition of the October image. This demonstrates that image data covering the entire growth period is valid for the identification of both crops without producing redundant information, which is closely related to their complex cropping structure and spatial distribution.

Additionally, considering the positive impact of optimal feature selection in some related researches, this study compared the classification results of this hierarchical approach

without further feature selection with that of "the backward feature elimination strategy" proposed by Hu et al. [48]. The comparison shows that only in the wheat identification system can the 12 features with a low ranking of importance be removed to ensure nearly the same overall accuracy, whereas in the autumn harvest crop recognition system, further feature selection did not have a positive effect on crop identification. This indicates that feature redundancy is not large enough in either classification model. Therefore, there is not a necessity for separate optimal feature selection in this study.

*4.2. Reliability of the Research Framework*

In this study, a framework for crop information identification was established to a generate crop rotation map based on the extraction of summer harvest crops and autumn harvest crops respectively. The successful implementation of this framework can be attributed to the preparation of high-quality remote sensing images collection, the combination of optimal features, applicability of the framework to complex cropping pattern, reliable algorithm, and the GEE platform's ability to map large spatial areas.

Firstly, Sentinel-2 not only has frequent revisit cycle and high spatial resolution, but also rich spectral information, with the three red-edge bands playing an important role in the identification of crops [12]. The 10-m resolution Sentinel-2 monthly time series images obtained by image pre-processing largely capture the key phenological periods for fragmented cropland and various crop types.

Secondly, the feature selection process was performed by progressively adding features to check the overall classification accuracy and F1-scores for crop identification over the entire growing season. The contribution of all features to the classification model was output after the classification was completed. As shown in Figure 13a, the top five features during wheat mapping were LSWI, NDWI, NDPI, NIR, and GI. It can be seen that features which are extremely sensitive to vegetation, water, and soil moisture (e.g., LSWI, NDWI) played more positive effects [49]. From Figure 13b, it was found that Elevation, NDWI, LSWI, NDVI, and NDRI played a crucial role in the mapping of the autumn harvest crops, especially the topographic factor Elevation. In northern China, cropland is more influenced by topographic factors, which may be related to the distribution of irrigation facilities, as precipitation in these areas is not as sufficient as in the south to meet the water need of double-cropping crops. Topographical features are certainly the crucial variables for the mapping of peanut, which is mostly grown in hilly areas.

Third, the framework for separate mapping of crops by growing season in this research was concerned with the serious problems that might be faced in directly identifying crops with information on crop rotation: (a) The cropping structure in Shandong is very complex, which makes it difficult to define the classes of training samples with crop rotation information, and the quantity and quality cannot be guaranteed; (b) mapping crops with crop rotation information is more likely to be faced with the problem of large intraclass variability and interclass similarity, e.g., where there is one season of the same crop in a double-cropping area, the interclass variability will be smaller for different crop rotation classes. In contrast, the framework proposed in this study addressed these problems well (Figure 14). The summer harvest and autumn harvest crops in the research were determined from the agricultural statistical information in the Provincial Statistical Yearbooks, which is authoritative. Then, based on the crop information provided by the China Agricultural Information Network, the sample dataset was produced by studying the unique phenological periods and other characteristics of various crops. Then, crops from different growing seasons were mapped separately to initially explore the rotation between the two seasons. Based on this, the temporal NDVI profiles and spatial distribution were analyzed to finally determine the rotation of major crops. Two separate crop extraction systems further reduced interclass similarity in the crop identification process and improved the accuracy of crop mapping. The crop rotation map produced in this research better identified fragmented crop plots under smallholder farming patterns, and the scattered villages, major roads, and rivers were well excluded, as shown in Figure 14.

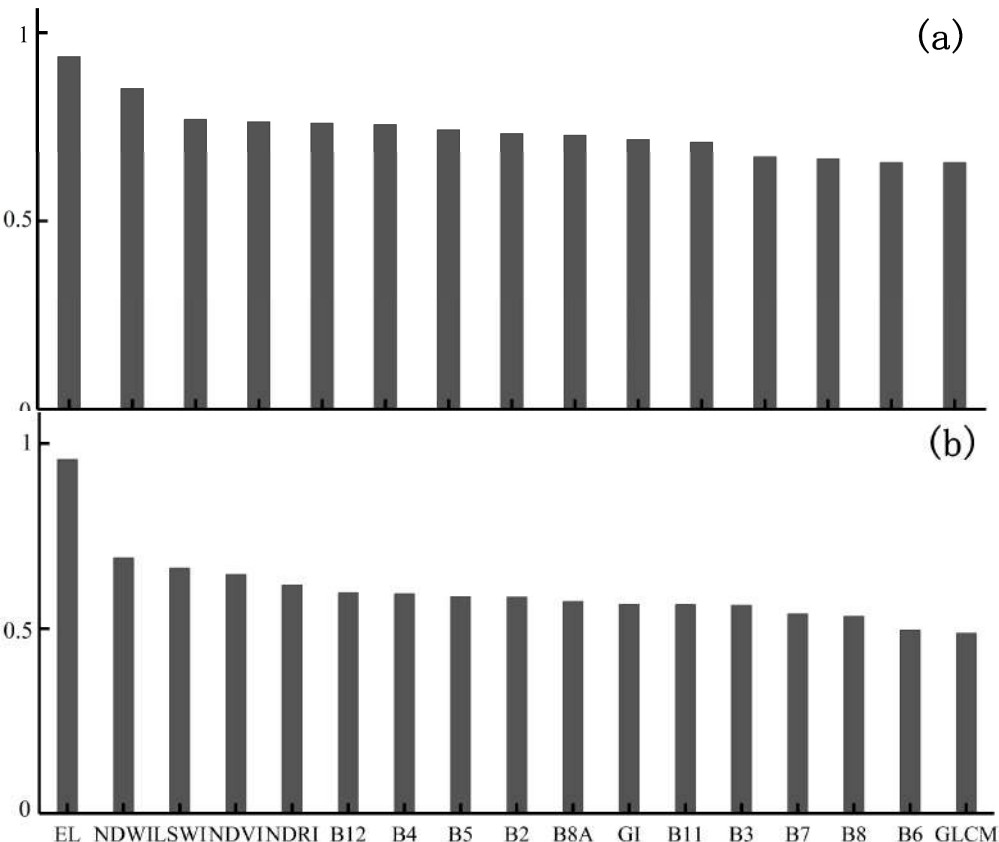

**Figure 13.** Average normalized importance of features in the identification schemes for summer harvest crops (**a**) and autumn harvest crops (**b**). Note: EL means the elevation, GLCM means the texture features.

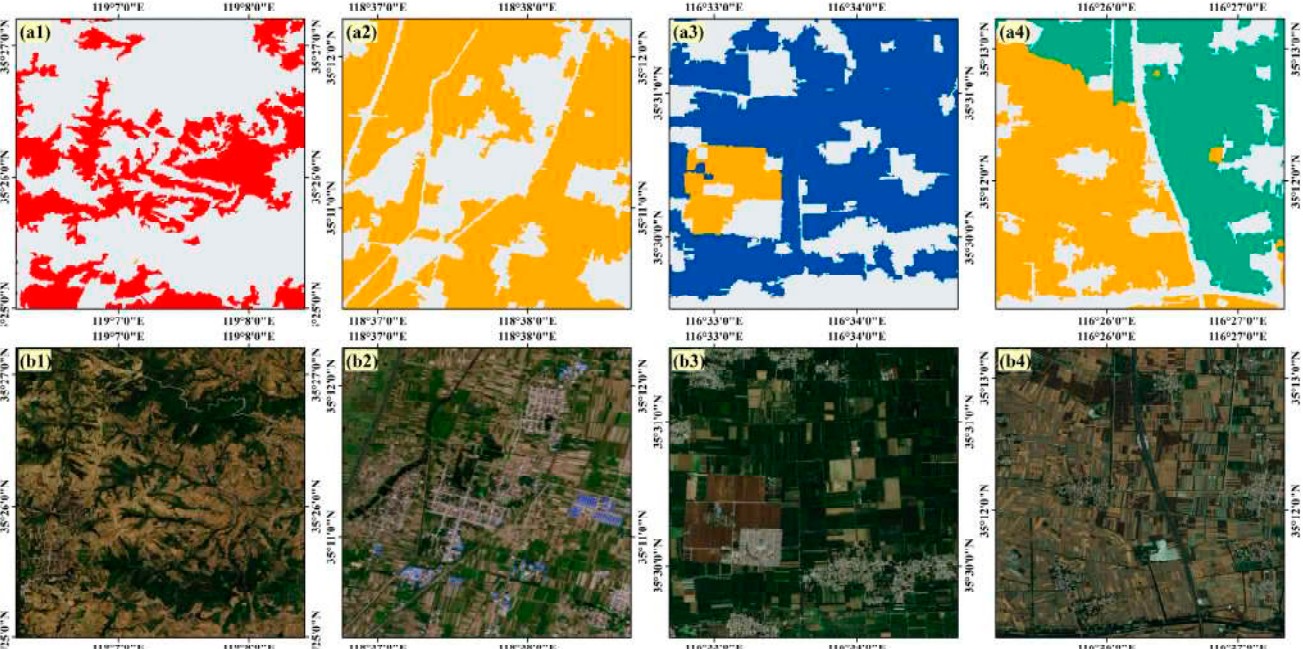

**Figure 14.** Crop rotation map generated in this research (**a1–a4**), and high spatial resolution satellite imagery for reference (**b1–b4**), from left to right are details of peanut, wheat–maize, cotton, and rice identification under smallholder agriculture.

Crop rotation types were defined on the basis of the crop rotation maps generated in this research and then mapped using the traditional method (direct mapping of crop rotation types). The mapping results of the traditional method were compared with the results of this research (Figure 15). The comparison shows that the crop rotation maps of this research are significantly better than those of the traditional method. As shown in these figures, the traditional method did not accurately identify crops under complex cropping patterns, producing serious under-classification, especially in cotton rotation and peanut growing areas. In the case of cotton rotation areas, where the first-season crop is complex and not the major crops, analysis of images throughout the crop year leads to much greater intraclass variability than analysis of images within the growing season of cotton, resulting in inaccurate mapping of cotton.

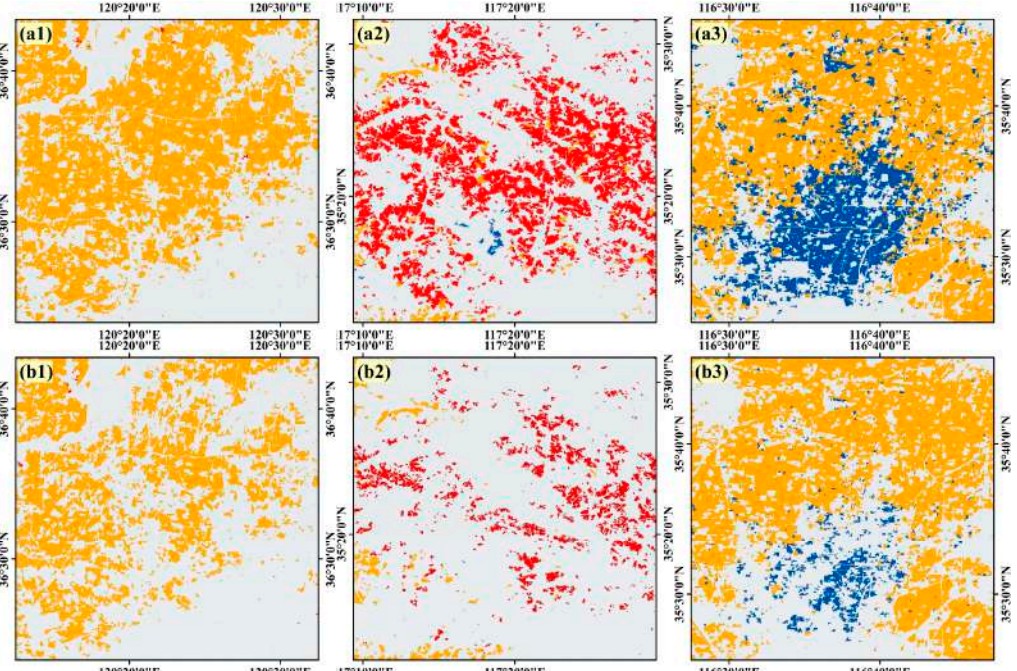

**Figure 15.** Comparison of the crop rotation maps generated in this research (**a1**–**a3**) with those generated by the traditional method (**b1**–**b3**), 1 to 3 showing wheat–maize, peanut, and cotton growing areas, respectively.

Finally, the robustness of the RF algorithm and the ability of GEE to handle large amounts of image data are also key factors in the successful implementation of this framework. The RF allows for both efficient supervised classification and also avoids overfitting when dealing with high-dimensional data. It has been widely used in land cover classification and has been proven to be effective. The processing and computing of several thousand scenes of imagery in this study is facilitated by the GEE cloud computing platform and the open availability of satellite data, which provide great convenience for the extraction of crop planting information.

### 4.3. Uncertainties

Shandong has a very large area of cropland, but the spatial distribution of cropland is very fragmented in some areas due to the topography [21,22]. Crop type identification in such areas faces serious mixed-pixels problems, even at 10 m resolution [50]. Furthermore, the variability of crop phenology and cropping systems over large areas will restrict the classification accuracy [4]. Different spatial locations, climatic conditions, management practices, etc. can even lead to high variability in the spectra and phenology of the same crops [51,52]. In addition, the predominance of smallholder systems has led to a high degree of complexity and flexibility in crop cultivation patterns [53]. Family-based

farming practices might lead to significant differences between the crops grown in one area and those grown in neighboring areas [52,54], and it is generally oriented by economic returns, with intercropping and other complex cropping patterns very common [12,54]. In the case of cotton, four cropping patterns exist: pure or intercropped cotton in single-cropping areas; seedling transplants or direct sowing cotton in double-cropping areas. This complex cropping pattern of a single crop increases the difficulty and uncertainty of crop rotation mapping.

Despite the successful implementation of crop rotation mapping in Shandong under a complex cropping system, there are still significant research possibilities due to the specificity of the crops mentioned above. Firstly, Shandong has a variety of crops, and only five main crops of wheat, maize, rice, peanut, and cotton were extracted for this research. For some crop types (soybean, tubers, millet, and other vegetables), they were not mentioned in this research because of the small acreage and the difficulty of producing high quality samples, and also to avoid imbalance with other major crops [55]. Future research could be carried out specifically on minor crops to further improve the accuracy of crop information identification. Secondly, in addition to crop type identification for single-cropping areas and crop rotation mapping for double-cropping areas, the cropping information extraction also includes intercropping mapping, etc. However, the presence of smaller cropland plots, irregular planting dates, the diversity of intercropping crops, as well as inter-crop similarities and intra-crop heterogeneity make intercropping monitoring more difficult [4,56]. Intercropping has not been studied in this research for now, but it is certainly a separate unit to be studied in the future. In addition, crop phenology varies widely across regions and agrometeorological conditions. For large-scale crop rotation mapping, subdivision into climatic or phenological zones can be tried to further reduce uncertainty.

## 5. Conclusions

This research established a sub-seasonal crop information identification framework for crop rotation mapping based on time series Sentinel-2 imagery of Shandong Province where both single-cropping and double-cropping crops exist. Firstly, Sentinel-2 data from the 2020 crop year were filtered and pre-processed to generate a monthly image dataset. Secondly, two different spectro-temporal feature combinations were generated to map the summer harvest crop and the autumn harvest crops, respectively. Thirdly, the classification results obtained by the pixel-based RF algorithm were optimized with the objects produced by image segmentation. Finally, a crop rotation map of Shandong was generated based on the independent generation of two crop maps.

The crop maps were evaluated using the validation samples. The summer harvest crop map had an OA of 0.93 with a Kappa coefficient of 0.86, while the autumn harvest crop map had an OA of 0.85 with a Kappa coefficient of 0.80. The mapping results show that crop rotation practice mainly occurs in the plains of western and central-eastern Shandong; the predominant crop rotation pattern is wheat and maize; rice is generally grown after the wheat harvest; peanut is mostly grown in a single-cropping region and is widespread and scattered; and cotton is grown on a relatively small area as a second season crop in rotation with other crops. In addition, LSWI, SAVI, etc. during the peak growing 'season achieved better performance in wheat identification, and elevation, NDVI, LSWI, GCVI, red-edge, and other spectral bands exhibited superiority in maize, rice, peanut, and cotton identification.

This research demonstrates the capability of the framework to identify crop rotation patterns and the potential of the multi-temporal Sentinel-2 for crops mapping under complex cropping systems. The framework is well transportable and can be applied to other years or to different cropping systems. The 10-m crop rotation map produced by this study could provide valuable information for cropland management, crop rotation monitoring, and agricultural policy development.



**Author Contributions:** Conceptualization, H.X. and B.C.; Methodology, H.X. and B.C.; Software, B.C.; Validation, B.C. and M.L.; Investigation, M.L.; Data curation, B.C.; Writing—original draft, H.X. and B.C.; Writing—review & editing, H.X. and M.L.; Visualization, B.C. All authors have read and agreed to the published version of the manuscript.

**Funding:** This research was jointly funded by the Shandong Provincial Natural Science Foundation (No. ZR2022YQ36), Open Fund of State Laboratory of Information Engineering in Surveying, Mapping, and Remote Sensing, Wuhan University (No. 20S01).

**Data Availability Statement:** The cropland maps generated in this research can be accessible from the corresponding author upon request.

**Acknowledgments:** We would like to thank the editor and the anonymous reviewer, whose constructive comments will help to improve the presentation of this paper.

**Conflicts of Interest:** The authors declare no conflict of interest.

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
