# Peer review of "A Sub-Seasonal Crop Information Identification Framework for Crop Rotation Mapping in Smallholder Farming Areas with Time Series Sentinel-2 Imagery"

_remotesensing, doi:10.3390/rs14246280_

Round 1
Reviewer 1 Report
The study proposed to classify crops in double cropping with crop rotation using two classifiers for two growing seasons in a year respectively.
The revision comments:
(1) Title is misleading. Crop rotation is not really what the project focuses on but classifying crops while double cropping is practiced with crop rotation in one year. The study only uses one year’s data to do the classification.
(2) The design of the study needs to be improved by extending the classification of cropland over multiple years if crop rotation mapping is the goal. The study claims to produce crop rotation map. However, the experiment only uses one year’s data. It only produces the crop classification maps of double cropping with crop rotation within a year. The framework of two crop classification systems in one year may be specifically applicable in the study area where only double cropping is practiced. The separation of crop growing seasons in a year does not consider actual growing seasons of different crops. The phonological information has actually been widely used in classifying crops. The arbitrary separation of a year into two seasons can be flawed which may actually degrade the features for classifying different crops.
(3) Revision suggestion is as follows if crop rotation mapping is still the goal:
a. Abandoning the two classification system framework. Instead, different crops may adopt different classification systems which match the actual growing season of crops. For example, winter wheat may use a specific classifier, soybean may use another classifier, and peanut may use one classifier. Start and end of crop growing season may be roughly decided by local crop calendar if seasons of satellite images are used separately. For the study area, the two season classification systems may end up at being actually used considering the specific characteristics (i.e. crops) of the study area. The classifier may actually use the phonological signature in classifying different crops. For example, planting dates for maize and soybean may be different, which can be effectively used a signature to distinguish these two crop types.
b. In the study case, multiple years of data should be used to produce the crop rotation map. This would show the performance of the classification methods for crop rotation mapping.
c. Other intensive cropping systems (e.g. intercropping) in one year may be discussed if the study is meant to develop a classification methodology for crop rotation mapping considering crop rotation within a year.
Reviewer 2 Report
HI Authors
Please check the comments on the attached documents and address the issues accordingly.
Regards
Reviewer

Reviewer 3 Report
This research paper aimed to investigate the crop rotation mapping. Overall, the paper was well-written and quite clear in its methodology. The authors presented adequate results and discussion for scientists in this matter, using it as a basis for future research and development.
Few suggestions to improve the work overall.
1. Reduce abstract length
2. L28-29: it is not these indexes that achieve better performance but they enable better performance
3. L39: insert link as citation and add reference at the bottom to facilitate reading.
4. L46-L48: can definitely be written better. It is not clear
5. L354: eliminate halo of graph label
6. L436: improve quality of graphs. Maybe reduce number of variables shown
7. L460: improve quality of figure. Enlarge maybe make not 2 rows and four columns but four rows and 2 columns
In addition, please check for typographical errors.
Round 2
Reviewer 1 Report
This revision reflect some of the suggested change as commented.